# Present and Theoretical Applications of Poly-Ether-Ether-Ketone (PEEK) in Orthodontics: A Scoping Review

**DOI:** 10.3390/ma15217414

**Published:** 2022-10-22

**Authors:** Tim A. P. Nai, Burcu Aydin, Henk S. Brand, Ronald E. G. Jonkman

**Affiliations:** 1Department of Orthodontics, Academic Center for Dentistry Amsterdam, Gustav Mahlerlaan 3004, 1081 LA Amsterdam, The Netherlands; 2Department of Oral Biochemistry, Academic Center for Dentistry Amsterdam, Gustav Mahlerlaan 3004, 1081 LA Amsterdam, The Netherlands

**Keywords:** orthodontics, poly-ether-ether-ketone, PEEK, materials science, nickel–titanium

## Abstract

**Background:** During the last decade, there has been an increased demand for non-metallic materials in orthodontics due to allergies, compatibility with medical imaging devices such as MRI, and aesthetic reasons. Monolithic poly-ether-ether-ketone material could address medical issues such as allergies and MRI compatibility. Moreover, nickel–titanium (NiTi) archwires covered in PEEK, either by a tube or electrophoretic deposition, could address esthetic concerns. This scoping review aims to summarize the available evidence in the literature to provide an overview of the applications and material properties of PEEK in orthodontics. **Methods:** This scoping review was conducted according to the Joanna Briggs Institute Manual for Evidence Synthesis for scoping reviews and the Preferred Reporting Items for Systematic Review and Meta-Analyses Protocols extension for Scoping Reviews (PRISMA-ScR). We searched for relevant publications in MEDLINE (via PubMed), Embase, Web of Science, Cochrane Library, CENTRAL, ProQuest, and SCOPUS. A gray literature search was conducted on Google Scholar. **Results:** Six studies were included. In three studies, the authors investigated the feasibility of developing a composite PEEK-NiTi wire, while in two other studies, the authors investigated the feasibility of monolithic PEEK wires. In the final study, the authors investigated the feasibility of PEEK as a bonded retainer. **Conclusions:** The included studies show promising results in developing monolithic and composite (PEEK-NiTi) materials. Further research on the robustness of PEEK composites in the oral cavity, the status of cytotoxicity and roughness values, and the (bio)-mechanical behavior of the composites is needed. A homogenously set up comparative study of clinically relevant, evenly sized, monolithic PEEK wires versus conventional orthodontic wires for their biomechanical, mechanical, and material properties would clarify the possibilities of developing monolithic PEEK wires. Missing data in the retainer study suggest more research on the mechanical properties and points of failure of PEEK-bonded retainers, and a comparative study comparing the failure and mechanical properties of PEEK-bonded retainers to flat braided metallic bonded retainers is needed.

## 1. Background

A growing interest in developing substitute materials with comparable mechanical characteristics to human bone, so-called “isoelastic” materials, started in the mid-1980s. During this time, researchers in the field of orthopedic surgery, who were looking for possible new biomaterials for use in spinal trauma and hip stems, became aware of the existence of polyaryletherketones (PAEKs) [1,2], a family of super engineering plastics (SEPs) or high-performance polymers (HPPs). The two most frequently used PAEK polymers in medicine are poly-ether-ether-ketone (PEEK) and polyether-ketone-ether-ketone-ketone (PEKEKK) [3].

Due to its chemical structure, PEEK is highly stable at low and high temperatures exceeding 300 °C and is resistant to chemical and radiation damage, allowing its sterilization prior to medical application. PEEK is compatible with reinforcing agents (such as carbon fiber, creating carbon fiber-reinforced PEEK, or CFR-PEEK), which enables tailoring of the elastic modulus [1]. Because carbon fiber-reinforced versions of PEEK have an elastic modulus of 18 GPa, comparable to human cortical bone [4], dental implantology researchers also became interested in this material. This isoelasticity makes PEEK a promising alternative in dental implantology with more favorable biomechanical properties than conventional dental implant materials. However, due to the limited number of studies (seven animal, one clinical) with PEEK, titanium remains the material of choice in dental implantology [5]. PEEK has also been investigated in adhesive dentistry as an alternative to materials such as zirconia and acrylics [6]. The adhesion of PEEK to composite resins was proven feasible, and a protocol was published [7].

There has been an increased demand in orthodontics for non-metallic materials during the last decade [8]. To improve esthetics, avoid metal allergies, and avoid interference with magnetic resonance imaging (MRI), glass fiber-reinforced resins were investigated as an alternative to metal archwires. However, glass fiber-reinforced resins exhibited considerable bending and distortion, disrupting the fiber–polymer interface and decreasing their mechanical properties [9,10]. While coating the wires with Teflon or polyethylene improved esthetics, problems with corrosion and metal allergies remained [11,12].

Several manufacturers have developed PEEK-coated wires in recent attempts to enhance the biocompatibility of NiTi wires. The PEEK coating was applied by either a tube covering the wire or electrophoretic deposition [13,14]. Electrophoretic deposition of a PEEK coating to nickel–titanium (NiTi) wires showed negligible microcracking of the coating. The data suggested electrophoretic deposition as a promising technique to manufacture more biocompatible wires [14]. The tube-covered NiTi wires were tested on color stability and frictional properties. Results showed comparable color values to conventional white-colored wires and reduced friction [13].

Several investigators evaluated the load-deflection and friction properties of orthodontic wires made from PEEK to assess their feasibility as a replacement for (elastic) NiTi wires [8,15]. They concluded that PEEK in a self-ligating system has comparable mechanical properties to NiTi wires and provides an esthetic, metal-free alternative to conventional orthodontic wires [15]. However, more research in a simulated in vivo environment is needed to determine its clinical performance before it can be suggested as an alternative [8,15]. Notably, the current state of research, development, and applications of PEEK in orthodontics has not been described in the literature. Thus, to map the current knowledge and provide a basis for future research, this scoping review of the literature on the material properties and applications of PEEK in the field of orthodontics was performed.

## 2. Methods

### 2.1. Eligibility Criteria

This scoping review protocol followed the Joanna Briggs Institute Manual for Evidence Synthesis for scoping reviews [16,17] and the PRISMA-ScR (Preferred Reporting Items for Systematic Review and Meta-Analyses Protocols extension for Scoping Reviews) [18]. The PICOS framework was applied to find relevant literature regarding the rationale (Table 1).

Included in this review is any study regarding the application of PEEK in orthodontics or fundamental research leading to possible applications in orthodontics, including in vivo studies and/or clinical trials. Exclusion criteria are the application of (part of) restorative treatment in dentistry, expert opinions or case reports/series, book chapters, papers published as manufacturers’ statements, and non-indexed journals.

### 2.2. Information Sources and Search Strategy

The search was conducted in May 2022. Studies published between 2010 and May 2022 were included using combinations of terms (Table 2) in MEDLINE (via PubMed), Embase, Web of Science, Cochrane Library, CENTRAL, ProQuest, ClinicalTrials.gov, and SCOPUS. Moreover, an additional search for publications indexed in other databases was conducted on Google Scholar. In line with the recommendations of Greenhalgh and Peacock [19], the reference lists of included studies were hand-searched to find potentially relevant references. Reference management was conducted in EndNote X9 (Thomson Reuters, Philadelphia, PA, USA).

### 2.3. Study Selection

Studies were selected in two phases. One reviewer (TN) searched the electronic database with a pre-determined syntax and set the search to titles, abstracts, and keywords when applicable. The process was documented, and a second reviewer (BA) re-ran the search to verify the results. Duplicates were removed by hand. Both reviewers performed TIAB screening on the unique records extracted from the primary search and assessed the eligible full-text records for inclusion. Differences in opinion concerning eligibility for inclusion were discussed between the reviewers. Figure 1 shows a schematic representation of the process.

Patients and the public were not involved in this research project’s design, conduct, reporting, or dissemination plans.

### 2.4. Data Extraction

Any relevant data concerning (bio)mechanical behavior, material properties, or biocompatibility was recorded.

### 2.5. Outcomes

The primary outcomes were defined as any relevant data concerning (bio)mechanical behavior, material properties, and biocompatibility. Secondary outcomes were not defined.

### 2.6. Data Synthesis

The high heterogeneity of the reported outcome measures did not allow the quantitative synthesis of the data retrieved from the included studies. Therefore, a qualitative synthesis of the results was performed.

## 3. Results

Of the 635 unique records, six studies were included in this review (Figure 1). Three studies investigated different approaches to a composite PEEK-NiTi wire [14,15,21]. Two studies investigated the feasibility of monolithic PEEK wires [8,16]. Kadhum and Alhuwaizi et al. investigated using PEEK as an orthodontic retainer wire [22]. Three studies were performed in Japan [8,15,16], two in Europe [14,21], and one in Iraq [22]. A summary of the included studies is shown in Table 3.

Boccaccini et al. assessed the feasibility of sintering a PEEK layer to NiTi wires [14]. Shape memory alloy (NiTi) wires were covered in PEEK or a combination of PEEK and PEEK/Bioglass using electrophoretic deposition (EPD). The best results were achieved with suspensions of PEEK powders in ethanol (1–6 wt%), using a deposition time of 5 min and an applied voltage of 20 volts. A homogeneous microstructure used these parameters along the wire length, and a uniform thickness of up to 15 μm was achieved without the development of cracks or the presence of large voids. PEEK powder was suspended in ethanol with the addition of Bioglass particles (0.5–2 wt%) (size < 5 μm) to produce PEEK/Bioglass coatings. Post-EPD, a sintering process with a final sintering temperature of 340 °C was applied to densify the coatings and improve the adhesion of the coatings to the substrate. The sintered coating retained high flexibility; hence, the wires could be strongly bent without flaking or cracking the PEEK coating. Moreover, the super-elastic properties of the NiTi wires were not negatively affected.

Sheiko et al. [21] assessed that the common problem of adding a polymer coating (such as PEEK) to NiTi alloys is the worsening of adhesion due to the alteration of the surface topography when phase transformation of the alloy to a martensitic state is induced. It led the investigators to a different approach by dip-coating the NiTi wire in PEEK. Sample preparation was performed by cleaning the wire by immersion in an ultrasonic bath, removing surface impurities by etching, obtaining an oxide layer with a mirror-like finish by electrochemical polishing (passivating), and finally dip-coating by immersing the substrate in a colloidal solution of PEEK (37 weight% in water). The coating had an average thickness of 12 μm.

The investigators showed a 10–100 fold smaller release of Ni-ions compared to passivated NiTi. Moreover, mechanical stress in continuous cycles of axial compression/stretching at 20% strain for 7 million cycles showed no substrate delamination. However, locally applied large pressures (>2 GPa) might fracture the PEEK film, which could lead to exposing the NiTi wire.

Shirakawa et al. assessed the feasibility of covering orthodontic archwires in PEEK tubes by sliding the tube over the wire (the measurements of the tubes were 0.5 × 0.6 μm or 0.8 × 0.9 μm, and the final thickness of the tubes covering the wire was 0.05 mm) [13]. Different archwires were used as substrates: 0.018 inch and 0.017 × 0.025 inch in stainless steel, cobalt–chromium, and nickel–titanium. The color of the PEEK wires was assessed by comparing them to a Vita A1 shade tab and conventional white-coated wires from Dentsply and Biodent. No significant difference in esthetics regarding the color shade was found.

Furthermore, friction and surface roughness was tested in an unspecified conventional bracket. Archwires were ligated in the bracket with an elastomeric module. Uncoated and PEEK-coated 0.018-inch archwires showed no difference in friction during sliding, while the PEEK-coated 0.017 × 0.025-inch archwires showed a 50% reduction in friction values compared to their uncoated counterparts. Analysis of the base surface of the bracket slot showed no surface irregularities (scratches) after sliding any of the PEEK-coated wires. However, the uncoated wires showed significant surface irregularities of the bracket slots after sliding.

Mechanical properties of wires fabricated from PEEK were investigated by Maekawa et al. [8], comparing PEEK plates (measuring 0.039 × 0.039 inch) to stainless steel, NiTi, TMA, and Co-Cr orthodontic archwires (measuring 0.016 × 0.022 inch). The wires were tested for load-deflection and bending-creep properties. The PEEK plates showed comparable load values characteristic of NiTi wires. Permanent deformation after the load-deflection test of the PEEK plates was negligible compared to NiTi wires. During the bending-creep test, a deformation of 2 mm was applied. Measurements (after 2 weeks and 2 months) showed a permanent deformation of the PEEK plates of 1 mm. The investigators did not include NiTi wires in the bending-creep test.

A comparison of mechanical characteristics of clinically relevant-sized PEEK wires (0.016 inch, 0.016 × 0.022 inch, 0.019 × 0.025 inch) to NiTi wires (0.016 inch) was described by Tada et al. [15]. The investigators determined load-deflection values and permanent deformation, load decrease in a stress-relaxation test, and static friction. The wires were fixed in a ceramic bracket of unknown dimensions with stainless steel covering the inside of the slot. The wire was fixated without ligation, elastomeric module ligation, or self-ligation. A three-point bending system was used to determine the load-deflection values.

The PEEK wires showed comparable load-deflection values to the 0.016-inch NiTi wire. Permanent deformation of an unspecified “greater degree” was observed in the PEEK wire compared to the NiTi wire. The investigators concluded the 0.019 × 0.025-inch PEEK wire to be the most comparable to the 0.016-inch NiTi wire when comparing the load values. In the stress-relaxation test, the wire was deflected at 2 mm and fixated for 24 h. The investigators determined the amount of stress (or force) exerted by the wire after 24 h. The NiTi 0.016-inch wire retained 78% of its initial force, while 0.016-inch PEEK, 0.016 × 0.022-inch PEEK, and 0.019 × 0.025-inch PEEK wires retained 77%, 67%, and 69% of their initial force, respectively. No statistically significant difference was found in the amount of static friction between the PEEK wires and the NiTi wire.

Kadhum and Alhuwaizi [22] investigated the efficacy of PEEK wires as a fixed orthodontic retention wire. The study’s objectives were to test the wires for bonding failure and pull-out strength. The investigators defined bonding failure as either failure of the wire or the bonding interface. PEEK wires measuring 0.0315 inch in diameter were used, with three types of bonding pre-treatment (no pre-treatment, air abrasion, and air abrasion/conditioning with methyl methacrylate monomer). The control groups consisted of a stainless-steel dead-soft coaxial wire (0.0195-inch), a stainless-steel three-stranded braided wire (0.010 × 0.028-inch), and a dead-soft solid flat titanium wire (0.010 × 0.028-inch).

In the bonding failure test, the investigators observed the comparable failure of the PEEK wires to the dead-soft metallic wires (both suffered from wire rupture). Only the braided wire was observed to fail at the bonding interface. The results of the pull-out performance test showed that the PEEK wires pre-treated by air abrasion or air abrasion/monomer conditioner suffered from wire rupture. Moreover, all metallic and non-treated PEEK wires were pulled out of the bonding interface.

## 4. Discussion

In recent years, the demand for non-metallic orthodontic appliances has increased, mainly to improve esthetics and avoid complications due to allergies and interferences with magnetic resonance imaging (MRI) [23,24]. Polymer-coated, white-colored wires showed significant improvement in esthetics [11,12]. However, after some time remaining in the oral cavity, the wires lost their coating, exposing the underlying metal [23,25]. They also did not have acceptable color stability [25], showed friction between the bracket and wire equivalent or higher than conventional systems [26], or had not solved problems with corrosion and metal allergies [11,12]. Uncoated non-metallic archwires exhibited considerable bending and distortion, which disrupted the fiber–polymer interface and decreased their mechanical properties [9,10]. The mechanical properties were so poor that they had a placebo function at best [20].

Thus, HPPs such as PEEK have allowed researchers and developers to apply new solutions to old problems. Two main approaches for using PEEK have been investigated: the development of composite wires and the development of monolithic wires as a substitute for metallic wires. In developing composites, the good mechanical properties of metallic materials (NiTi) are combined with the biocompatible and esthetic properties of non-metallic materials (PEEK). In contrast, monolithic wires are fabricated solely from PEEK and are supposed to act as a substitute for NiTi wires.

Boccaccini and Sheiko investigated the development of composites by coating NiTi wires with PEEK. Moreover, Shirakawa investigated the feasibility of covering NiTi wires with a PEEK sleeve (sleeving), while Boccaccini et al. developed a PEEK-NiTi composite using electrophoretic deposition (EPD). To improve adherence of the coating (PEEK) to the substrate (NiTi) after the EPD process, the investigators applied a sintering process with a temperature of 340 °C for 20 min [14]. Heating NiTi up to 300 °C activates the shape memory effect and restores its original shape [27]. At temperatures of approximately 540 °C, the crystalline structure will reorient itself, causing permanent shape deformation in an annealing process. Heating the NiTi wire between 400 °C and 500 °C for prolonged periods (60 min) causes deterioration of its super-elastic properties [28].

Dip-coating, as used by Sheiko et al., could solve the potential deterioration adhesion of the coating to the substrate by not having any adhesion, allowing the coating to flex with the substrate in its martensitic state. Electrochemical polishing and dip-coating a NiTi wire could result in a 10–100-fold decrease in Ni-ion release compared to passivated or chemically etched NiTi [21]. Data on Ni-ion release from the EPD-coated substrate are unavailable, hence an opportunity for future research. Having a uniform coating thickness of 12 μm for dip-coating and up to 15 μm for electrophoretic deposition could imply that both coatings would not negatively affect NiTi’s super-elastic properties [14]. Surface roughness of dip-coated substrates on SEM imaging seems less compared to EPD-coated substrates; however, further research is needed to determine the mechanical and clinical relevance [14,21].

Covering over-the-counter orthodontic archwires with a PEEK tube (by sliding the tube over the wire) has shown to be a feasible and cost-effective way to fabricate an esthetically pleasing archwire [13]. Its frictional properties are favorable compared to non-coated wires [13]. The influence of the thickness of the PEEK tube on the wire dimensions, and thus its ability to fit in the bracket slot or ability to express torque, is unknown. Further research could provide clarification.

Another (non-composite) approach has also been investigated, fabricating orthodontic wires solely from PEEK or monolithic PEEK wires [8,15]. According to the authors, monolithic PEEK wires exhibited the same friction values as metallic archwires. However, a round, 0.016-inch NiTi wire was compared to 0.016 × 0.022-inch and 0.019 × 0.025-inch PEEK wires in a slot of unknown dimensions. PEEK wires showed permanent deformation of 50% of the initial deformation after 2 weeks and a loss of up to 31% of their initial exerted force after 24 h of deformation [8,15]. Thus, this rapid degradation of exerted force and permanent deformation is cause for concern when applying monolithic PEEK wires as in orthodontic treatment.

In contrast, applying PEEK in the passive (or less-active stage) of orthodontic treatment as a bonded retainer has shown favorable results [22]. The bonding procedure of PEEK retainer wires is comparable to metallic retainer wires, which is favorable for clinical logistics, and the mechanical performance of the PEEK retainer wire was shown to be comparable to that of metallic dead-soft bonded retainers. However, all of the PEEK retainer wires suffered from wire rupture during bonding failure tests. Due to the material properties of dead-soft materials, they are unable to withstand externally exhibited forces, frequently suffer from breakage, and are mainly fit for temporary usage [29,30,31]. Hence, caution is advised when applying PEEK retainer wires until more data is available.

The included studies show promising results in developing monolithic and composite (PEEK-NiTi) materials. However, several questions have remained unanswered and provide a basis for future research. For instance, it is unknown: how robust the PEEK-NiTi composites are in the oral cavity; the status of cytotoxicity and roughness values; and the (bio)-mechanical behavior of the composites. Thus, a homogenously set up comparative study of clinically relevant, evenly sized, monolithic PEEK wires versus conventional orthodontic wires for their biomechanical, mechanical, and material properties would clarify the opportunities in developing monolithic PEEK wires. Missing data in the retainer study suggest the need for more research on the mechanical properties and points of failure of PEEK-bonded retainers, and a comparative study of the failure and mechanical properties of PEEK-bonded retainers to flat braided metallic bonded retainers is needed.

By design, the goal of this review was to scope a wide body of literature to identify knowledge gaps. The shortcomings of this study are found in its design; only studies published in indexed journals were included. As a consequence, a number of studies not published in indexed journals were excluded [32,33,34].

## 5. Conclusions

The development of monolithic and composite PEEK materials shows promising results. Mapping the available evidence about the applications of PEEK polymers in orthodontics, we identified several opportunities for future research:Roughness values, robustness in the oral cavity, cytotoxic values, and (bio)-mechanical behaviors of (PEEK)-coated NiTi wires;Homogenously set up comparative studies, comparing clinically relevant-sized PEEK wires to conventional orthodontic wires;Mechanical properties and points of failure in PEEK-bonded retainers compared to braided flat metallic retainer wires.

## Figures and Tables

**Figure 1 materials-15-07414-f001:**
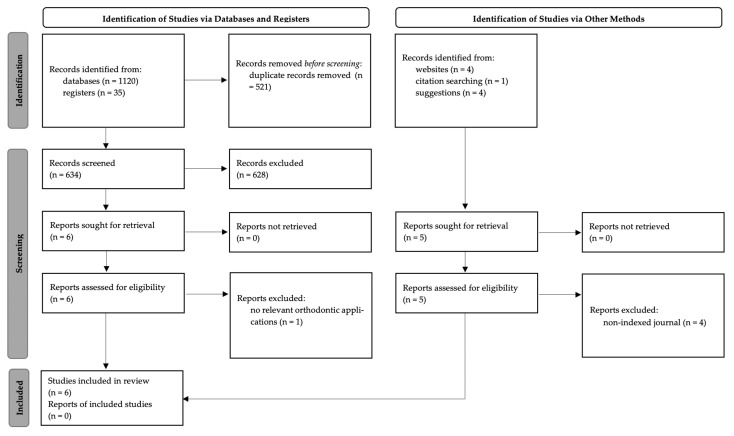
PRISMA flowchart [20] showing the study selection process.

**Table 1 materials-15-07414-t001:** Specification of the PICOS framework.

Population/Problem	PEEK Specimens
intervention	application as orthodontic technology, composite application with conventional orthodontic technology
comparison	conventional orthodontic technology
outcomes	biomechanical properties, material properties, biocompatibility
study design	any relevant study published on the use of PEEK as orthodontic technology

**Table 2 materials-15-07414-t002:** Pre-determined search syntax.

PEEK; poly-ether-ether-ketone; PAEK; polyaryletherketone; super engineering plastic; high-performance polymer; dentistry; orthodontics; wire; esthetic orthodontic wire; NiTi; Ni-Ti; nitinol; nickel–titanium; retainer; orthodontic retainer; OR #1/#6; OR #7/#16; #15 AND #16.

**Table 3 materials-15-07414-t003:** Main properties of included studies.

Author	Year	Country	Title	Objective of the Study	Outcome
Boccaccini et al. [14]	2006	The United Kingdom, Germany	Electrophoretic Deposition of PEEK and PEEK/Bioglass (R) Coatings on NiTi Shape Memory Alloy Wires	Feasibility assessment of covering NiTi wires with PEEK using electrophoretic deposition	The wires could be strongly bent without flaking or cracking the PEEK coating. The super-elastic properties of the NiTi wires were not negatively affected.
Kadhum and Alhuwaizi [22]	2021	Iraq	The Efficacy of PEEK Wire as a Retainer Following Orthodontic Treatment	Assessment of the efficacy of PEEK as an orthodontic retention wire	PEEK wires (0.0315-inch diameter) were compared to metallic controls. In the bonding failure test, comparable failure of the PEEK wires to dead-soft metallic wires was observed (wire rupture). The pull-out performance test showed that pre-treated PEEK wires (by air abrasion or air abrasion combined with a monomer conditioner) suffered from wire rupture. All metallic and non-treated PEEK wires were pulled out of the bonding interface.
Maekawa et al. [8]	2015	Japan	Mechanical Properties of Orthodontic Wires Made of Super Engineering Plastic	Assessing mechanical properties of clinically non-relevant-sized PEEK wires	PEEK plates measuring 1.0 × 1.0 × 200.0 mm were compared to NiTi archwires measuring 0.40 × 0.55 mm (0.016 × 0.022-inch). The PEEK plates showed load values and characteristics comparable to NiTi wires.
Sheiko et al. [21]	2016	France	PEEK-Coated Nitinol Wire: Film Stability for Biocompatibility Applications	Feasibility assessment of covering NiTi wires with PEEK using dip-coating	Mechanical stress in continuous cycles of axial compression/stretching showed no substrate delamination. Locally applied large pressures caused fractures in the PEEK film, leading to the exposure of the NiTi wire.
Shirakawa et al. [13]	2018	Japan	Mechanical Properties of Orthodontic Wires Covered With a PEEK Tube	Feasibility assessment of “sleeving” NiTi wires with PEEK tubes	PEEK-coated 0.017 × 0.025-inch archwires showed a 50% decrease in friction values compared to uncoated wires. The base surface of bracket slots showed no scratches after sliding PEEK-coated wires, while uncoated wires showed significant surface irregularities.
Tada et al. [15]	2017	Japan	Load-Deflection and Friction Properties of PEEK Wires as Alternative to Orthodontic Wires	Assessing mechanical properties of clinically relevant-sized PEEK wires	Clinically relevant-sized PEEK wires (0.016 inch, 0.016 × 0.022 inch, 0.019 × 0.025 inch) to NiTi wires (0.016 inch; round) were compared. The PEEK wires showed comparable load-deflection values to the 0.016-inch NiTi wire. Permanent deformation of an unspecified “greater degree” was observed in the PEEK wires compared to the NiTi wire. The 0.019 × 0.025-inch PEEK wire was most comparable to the 0.016-inch NiTi wire.

## Data Availability

All the data is available within the manuscript.

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
