# Peer review of "Present and Theoretical Applications of Poly-Ether-Ether-Ketone (PEEK) in Orthodontics: A Scoping Review"

_materials, 2022, doi:10.3390/ma15217414_

Round 1

Reviewer 1 Report

The authors presented review article based on application of PEEK as an orthodontic appliances.

Comments need to be addressed before taking any recommendation.

This review article is present on OSF sinch March 2022. Is it already link with Materials MDPI?

In the introduction, the authors provided a literature search on use of glass fibers in orthodontics. However, information related to their applications as the orthodontic retainer is missing. Include this with relevant information.

Methodology: A combination of keywords should be used, if used then provide that information in table.

The authors need to search again. Some data from the literature is missing such as:

https://jbcd.uobaghdad.edu.iq/index.php/jbcd/article/view/3147

The search should be comprehensive.

As authors found only 6 articles, I would strongly suggest include in vivo, clinical trials, case reports should be included in this review article, otherwise, very less interesting for readers.

Should article of Boccaccini should be included? as their focus was on coating of bioactive glass on PEEK and they suggested the application for biomedical purposes, not very specifically to orthodontics. I don't think this article should be included. However, you can mention this in Discussion and compare with other retainers such as glass fibers were coated/grafted with nano-apatite and evaluated the bond strength.

Reviewer 2 Report

The methodology presents some lacks and weaknesses.

The English language is often distracting and poor including orthographic errors.

TITLE

 It does not describe the intent of the work.

ABSTRACT

The aim should be better described and above all, it should emerge from the shortcomings found on the subject in the literature

The MATERIALS AND METHODS section has some lacks:

Please move the Table 2 as supplementary data table

The Prisma Flow chart is not the last one of 2021. Please update it

Please

DISCUSSION

This section should be better developed. It seems just a repetition of the results, only with a description of the data.

REFERENCES

are limited to those necessary to support the study but there are some typographical mistakes such as missing volumes or pages and punctuation marks.

Reviewer 3 Report

The article is interesting. It provides valuable up-to-date scientific information on new technologies used in orthodontics.

The article presents information on the use of PEEK in orthodontics on the basis of a literature review. The article is relevant, interesting and original. The paper is well written. The text is clear to read. The conclusions are consistent with the evidence and arguments presented.

Round 2

Reviewer 1 Report

As, the authors mentioned that the paper is no more available on OSF, however, I read full article over there. Double-check this.

The authors should clearly mentioned non-indexed journals e.g. not indexed in Scopus, WoS, PubMed etc.

Inclusion criteria shows search from the gray literature, Google Scholar then articles published in Journal of Baghdad College of Dentistry or other journals available on Google Scholar/Google should be included. Search Selection criteria needs to be improved.

Agreed with the authors that they don't want to include case reports/series/book chapters, manufacturer's statemtents. However, in vivo studies and/or clinical trials should be included. There is a difference between clinical trial and case report.

After taking the copyright permissions, the authors should include some figures to make this review more interesting, specifically morphological images.

Author Response

Dear reviewer, please see our reply to your comments in the  attachment

Round 3

Reviewer 1 Report

No further comments.